# Performance Analysis of a Vertical FSO Link with Energy Harvesting Strategy

**DOI:** 10.3390/s22155684

**Published:** 2022-07-29

**Authors:** Carmen Álvarez-Roa, María Álvarez-Roa, Francisco J. Martín-Vega, Miguel Castillo-Vázquez, Thiago Raddo, Antonio Jurado-Navas

**Affiliations:** 1University Institute of Telecommunication Research (TELMA), University of Málaga, CEI Andalucía Tech., E-29071 Málaga, Spain; calvarez@ic.uma.es (C.Á.-R.); m.alvarez@ic.uma.es (M.Á.-R.); fjmvega@ic.uma.es (F.J.M.-V.); miguelc@ic.uma.es (M.C.-V.); thiago@ic.uma.es (T.R.); 2Engineering, Modeling & Applied Social Sciences Center, Federal University of ABC, Avenida dos Estados 5001, Santo André 09210-580, SP, Brazil

**Keywords:** BER, vertical link, pointing errors, FSO, OOK, energy harvesting, atmospheric turbulence

## Abstract

In this paper we investigate the application of free space optical (FSO) communications, energy harvesting, and unmanned aerial vehicles (UAVs) as key technology enablers of a cost-efficient backhaul/fronthaul framework for 5G and beyond (5G+) networks. This novel approach is motivated by several facts. First, the UAVs, acting as relay nodes, represent an easy-to-deploy and adaptive network that can provide line-of-sight between the base stations and the gateways connected to the core network. Second, FSO communications offer high data rates between the UAVs and the network nodes, while avoiding any potential interference with the 5G radio access networks. Third, energy harvesting in the optical domain has the potential to extend the UAVs’ battery life. Nevertheless, the presence of atmospheric turbulence, atmospheric attenuation, and pointing errors in the FSO links severely degrades their performance. For this reason an accurate yet tractable modelling framework is required to fully understand whether an UAV-FSO backhaul/fronthaul network with energy harvesting can be applied. To this end, we consider a composite channel attenuation model that includes the effect of turbulence fading, pointing errors, and atmospheric attenuation. Using this model, we derive analytical closed-form expressions of the average harvested energy as a function of the FSO link parameters. These expressions can be used to improve energy harvesting efficiency in FSO link design. We have applied our proposed expressions to evaluate the energy harvested in vertical FSO links for a variety of real scenarios under a modified on-off keying (OOK) scheme optimized for energy harvesting. From the simulations carried out in this paper, we demonstrate that significant values of harvested energy can be obtained. Such performance enhancement can complement the existing deployment charging stations.

## 1. Introduction

### 1.1. Overview of FSO Communications

The combination of the high bandwidth of optical communications with the flexibility of wireless technologies offered by free-space optical (FSO) communications has led to a fresh wave of innovation and research activities in this field [1]. Aside from the higher bandwidth, FSO communications offer several advantages over classical RF based wireless communications. First of all, the extremely high carrier frequencies inherent to the optical links make FSO detectors immune to multipath fading which severely degrades the performance of RF links [2,3]. Furthermore, FSO technology has the potential to reduce cost and consumed energy. The spectrum above 300 GHz is unlicensed, so operators do not have to pay for exclusive access in optical bands. Moreover, the components used in FSO links are cheaper, smaller, lighter and have lower power consumption as compared to that of RF components [4]. More importantly, FSO technology does not interfere with RF systems, which paves the way to mixed RF/FSO approaches where the optical links complement the existing RF infrastructure [5,6]. Due to all these benefits, FSO communications have found their place in diverse applications such as space communications [7,8,9], underwater communications [10,11,12,13], indoor local area networks [14], data center networks [15], and mobile backhaul [6,16,17].

However, wireless optical links are affected by many impairments that may compromise their performance, so an accurate channel modeling is needed to anticipate the potential benefits of FSO-based approaches. These impairments are mainly categorized into four different effects: namely, pointing errors (i.e., misalignment losses), atmospheric losses, atmospheric turbulence (also called scintillation), and noise [1].

Thus, pointing errors are related to the misalignment of the transmit beam with respect to the field of view of the receiver. For fixed links of a few hundred meters, increasing the beam divergence can alleviate the misalignment loss at the expense of a higher geometric loss. Nevertheless, longer link distances or links with moving nodes require appropriate pointing, acquisition, and tracking procedures to mitigate the adverse effect caused by pointing errors [18].

Next, atmospheric loss is due to the presence of particles that either absorb or scatter the transmitted light (in that latter case, only the fraction of light scattered out of the location of the receiver is considered). In particular, the particles that affect FSO communication systems include those ones associated with rain, snow, fog, pollution, and smoke, among others.

On the other hand, the atmospheric turbulence or scintillation, is explained by inhomogeneities in the pressure and the temperature of the atmosphere that induce variations of the air refractive index along the transmission path [19]. These fluctuations cause random variations in the amplitude and phase of the received signal, i.e., fading, that lead to a considerable degradation in long-distance links. Several statistical models have been proposed in the literature to fully characterize the turbulence fading. The log-normal [20] distribution has been accepted as an accurate model for weak turbulence conditions whereas negative exponential [21] and Rayleigh [22] distributions have been used to model strong turbulence. For this reason, there have been remarkable research efforts to establish a common statistical model to characterize any turbulence condition. In this context the Gamma-Gamma distribution represents an appealing model that allows tractable analysis of the link performance while modelling turbulence conditions ranging from weak to strong turbulence [21].

Finally, the last adverse effect in FSO links is produced by noise. Background noise is the dominant one in most optical links. This noise is present due to the fact that the receiver collects some undesirable radiations such as reflected or scattered sunlight from hydrometeors or other objects. This radiation is mitigated by means of narrow spectral bandpass and spatial filtering prior to the photo-detector (PD). However, a non-negligible background noise might fall within the spatial and frequency ranges of the detector causing a random electrical signal that is added to the desired signal. This noise term can be modeled according to a Poisson distribution [23]; nevertheless, when the number of received photons associated with this background radiation is high enough, the Poisson distribution can be approximated by a Gaussian distribution [24]. These facts motivate the inclusion of an additive white Gaussian noise (AWGN) model in FSO links.

### 1.2. Related Work

The advances in pointing, acquisition, and tracking [18] have enabled the application of FSO communication to unmanned aerial vehicles (UAVs), which can act as relay nodes in 5G and beyond (5G+) cellular networks. This approach is especially promising in the context of backhaul/fronthaul networking [25] since the mobility of UAVs and the height at which they operate provides a reliable line-of-sight link for a high-bandwidth FSO connection [6]. In addition, the location of the relay nodes can be changed, making this kind of network adaptable to changes in weather and traffic needs. For instance, if any base station (BS) is highly loaded, UAVs can connect to that BS to readjust the backhaul traffic. Accordingly, if the atmospheric loss conditions worsen, e.g., due to fog, UAVs could approach the BSs to maintain a reliable FSO data connection.

However, one of the main drawbacks associated with extending the network with aerial access points is related to the service interruption when UAVs land to recharge their batteries. UAVs need to fly back to a nearby charging station to recharge their onboard battery frequently. For this reason, energy-efficient frameworks are preferred to extend the service time provided by UAVs. To this end, two different solutions (or a combination of both) are normally proposed: (i) trajectory optimization; and (ii) energy harvesting. The former approach consists on a route design to minimize the consumed energy while guaranteeing a target rate with a given node [26,27]. On another note, energy harvesting (EH) involves capturing, and storing energy from external sources, e.g., solar power, thermal energy, wind energy, or kinetic energy, which is generally known as ambient energy [28]. Interestingly, we can remark on the simultaneous lightwave information and power transfer (SLIPT), which is a kind of EH, where the captured energy comes from the optical signal that carries the information [29]. This latter approach is very promising since the optical signal uses narrower beams, and thus the emitted energy is concentrated towards the receiver, which makes SLIPT systems particularly efficient. Despite their relevant and potential benefits to extend the UAVs battery life, the number of works focused on the application of SLIPT for FSO-based UAVs nodes is limited. For the sake of clarity, some of these works are described below.

To start with, in [30] a two hop, mixed FSO-RF relaying scheme is proposed and analyzed assuming Gamma-Gamma turbulence fading and pointing errors. Under these assumptions, the outage probability in terms of the bivariate Fox-H function is derived. The proposed scheme considers a first hop in the optical domain where the relay can capture the energy from the received light. Then, this energy is used for RF signal transmission. Results reveal that a larger PD responsivity results in a better performance.

A pioneering work about the application of FSO with EH to UAVs-based networks for 5G backhauling is detailed in [6], and assessed via simulations in terms of capacity and cost. Simulation results are obtained for various different conditions involving atmospheric loss, turbulence loss, and pointing errors. Further investigation was performed in [31] where the SLIPT scheme was posited as an optimization problem to maximize the harvested energy while guaranteeing a target rate, for different typical modulations such as pulse amplitude modulation (PAM) and pulse position modulation (PPM).

A novel view about the SLIPT communication between BSs and UAVs nodes was presented in [32]. This work relies on mathematical tools from stochastic geometry to analyze the performance of UAVs based networks. With this approach, the main metrics are obtained as spatial averages over infinite realizations involving different BSs and UAVs locations. To assess the effectiveness of EH and FSO based communication, the joint energy and the signal-to-noise ratio (SNR) coverage probability are derived, i.e., the joint probability of the UAV receiving enough energy to ensure successful operation (hovering and communication) and having a received SNR higher than a given threshold.

To finish this brief review, we must cite the work presented in [33] where a mixed FSO/RF network with EH was proposed. In this scheme, ground stations transmit backhaul traffic to the UAVs, which act as moving BSs. Subsequently, the UAVs transmit RF signals to the ground users. Accordingly, the UAVs harvest and store the received energy coming from the ground stations through their FSO links. The proposed framework is considered as an optimization problem to maximize the energy efficiency.

### 1.3. Contributions

Despite their relevance, none of the aforementioned studies, e.g., [6,29,30,31,32,33], consider two important aspects: (i) the random nature of the EH, which is inherited from the variability of the optical channel; and (ii) the trade-off between bit error rate (BER) and EH. In this paper, we investigate the influence of this random behavior on the design of EHoptimized links, while fulfilling a target BER. This will be achieved by transmitting a certain fraction, ζ, of the peak transmission power, $P_{m}$, with the logical symbol ‘0’. As shown in this paper, a proper design of the ζ maximizes the harvested energy while fulfilling a desirable reliability in terms of the BER.

More specifically, we aim to respond to the following inquiries: (i) the optimum ζ value to maximize the harvested energy while fulfilling a target BER; and (ii) the maximum average harvested energy that can be achieved under realistic link conditions. To address those two issues we derive some closed-form expressions for the two main metrics here considered: the average BER and the average harvested energy. Therefore, the paper’s contributions can be summarized as follows:

1.We propose an accurate mathematical model for the UAV-based approach to provide backhaul connectivity for 5G+ networks [6]. To this end, ground stations and UAVs establish FSO links with EH to recharge the UAVs batteries and extend the service time. Furthermore, the UAVs will have another FSO link with the BSs to provide backhaul connectivity. The mathematical model considers realistic channel impairments such as turbulence fading (scintillation), atmospheric loss, pointing errors, and additive white Gaussian noise (AWGN).2.We derive analytical closed-form expressions of the average harvested energy as a function of the FSO link parameters. These expressions can be used to improve energy harvesting efficiency of the FSO links.3.Closed-form expressions for the BER have been obtained for links using a variant of the on-off Keying (OOK) scheme that has been properly modified to maximize the EH. With this scheme, which is referred to as OOK-EH, a certain fraction, ζ, of the peak optical power, $P_{m}$, is also transmitted with the symbol ‘0’ to recharge the batteries of the UAV.4.We obtain the optimal ζ value that maximizes the average harvested energy while maintaining its associated BER smaller than a *target* BER. Such a target BER is defined in practical communication standards to guarantee a reliable transmission.5.The performance associated with the proposed scheme is corroborated with extensive Monte Carlo (MC) simulations in diverse and realistic atmospheric conditions. An excellent agreement between theoretical and simulation results is observed. Results reveal that the combination of FSO, EH, and UAVs provide reliable backhaul links while capturing energy to extend the UAVs service time. Let us recall that short lifetime of the batteries mounted on drones is seen as the main limitation of the proposed approach. For this reason, one of the main targets of our work is to obtain the optimal amount of energy that can be collected to charge the batteries of the UAVs involved in the system, as discussed in [30,31], while maintaining its performance in terms of BER.

The rest of the paper is structured as follows. In Section 2 the system model is mathematically described, including the signal as the channel model. Then, the main metrics are presented and derived in Section 3, namely, the expressions for the amount of harvested energy by the system and its BER. In Section 4 numerical results are obtained to assess the benefits of the proposed scheme, which allows to identify both trends and limiting factors. Finally, relevant conclusions are discussed in Section 5.

## 2. System Model

### 2.1. Proposed Scenario

We consider a flying backhaul/fronthaul network composed of UAVs that provides reliable connectivity to a 5G+ radio access network (RAN). The UAVs act as networked flying platform (NFP) nodes that can react to changes in weather or traffic conditions [34]. Its ground-to-air links are based on FSO technology with EH. This solution has four remarkable benefits: (i) its FSO links provide the required high data rates for the backhaul or fronthaul connections; (ii) FSO links do not cause interference to the 5G RAN; (iii) UAVs can adapt their location to the traffic and channel conditions; and (iv) EH extends the UAVs’ service time and thus, improves the network performance.

This approach is illustrated with Figure 1 where it shows the representative nodes and connections. The backhaul traffic represents the IP data transmissions between the 5G core network and the BSs in distributed RAN approaches [35]. Accordingly, the BSs fully implement the 5G RAN protocol stack, and thus they are source/destination of IP packets towards/from the core network.

A recent approach to the classical distributed RAN arises with the cloud RAN (C-RAN) [36], where a base band unit (BBU) centralizes the RAN processing of many small cells. The signal transmission related to each cell is carried out by radio remote heads (RRHs) that are connected with the BBUs through fronthaul links. This latter approach offers two main benefits compared to distributed RAN. First, C-RAN reduces costs since each small cell can be implemented with a RRH, cheaper than a complete BSs. Second, C-RANs ease the implementation of coordinated mechanisms for interference mitigation.

Coming back to Figure 1, it is straightforward to see the different backhaul links between the core network and the BSs through the FSO-UAVs based network. This scheme requires a FSO gateway that creates the link with a mother NFP node, i.e., a mother UAV that forwards the IP traffic towards the NFP node currently connected to the target BS. Analogously, the fronthaul links between the BBU and the RRHs also require a connection between a FSO gateway and the mother NFP node. Certainly, many applications beyond classical mobile networks can benefit from the proposed scenario shown in Figure 1 since it can provide backhaul/fronthaul connectivity to 5G V2X, or even future underwater communications [11,12,13,37].

### 2.2. Received Signal Model

Throughout this paper, we consider a non-coherent intensity modulation with directdetection (IM/DD) scheme since it is a practical and widely extended technique for FSO communications [38]. With this scheme, the intensity of the emitted light is employed to convey the information. Therefore, the PD output current can be expressed as follows [39,40]
(1)i=hRx+n,
where *x* is the transmitted intensity in W, *R* is the detector responsivity in W/A, *h* is the channel coefficient and *n* represents the noise term at the receiver side. We assume that the shot noise induced by the ambient light is the dominant noise source in our analysis. Thus, we model *n* as a signal-independent zero-mean Gaussian noise with variance given by
(2)$$\sigma_{n}^{2}=2qRB_{n}P_{amb},$$
where *q* denotes the electron charge, $B_{n}$ is the receiver noise bandwidth, and $P_{amb}$ is the ambient light power. This latter term can be obtained as $P_{amb}=\pi r^{2}S_{n}B_{o}\Omega_{FOV}$, where *r* is the receiver aperture radius, $S_{n}$ is the ambient light spectral radiance, $B_{opt}$ is the optical bandwidth in nm and $\Omega_{FOV}$ is the receiver field of view (FOV) in srad.

The transmitted intensity is taken as symbols drawn with equal probability from the OOK constellation such that $x\in \{P_{1},P_{0}\}$, where $P_{1}$ and $P_{0}$ represent the power corresponding to the transmission of a ‘1’ and a ‘0’, respectively.

In this respect, we can cite three primary atmospheric phenomena that affect optical wave propagation and that constitute the total channel coefficient *h*: (i) the deterministic path loss, $h_{l}$ characterized by absorption and scattering; (ii) the geometric spread and pointing errors $h_{p}$; and (iii) the refractive-index fluctuations (i.e., the atmospheric turbulence), $h_{a}$, leading to irradiance fluctuations. Thus, the total channel attenuation is modeled as the product of these aforementioned channel factors as:(3)$$h=h_{l}h_{p}h_{a},$$
where $h_{p}$ and $h_{a}$ are random variables (RVs). In the next sections, we describe the underlying distribution that model those RVs. Finally, we define the signal to noise ratio as [41]:(4)$$\mathrm{SNR}=\frac{{\left(RhP_{av}\right)}^{2}}{\sigma_{n}^{2}},$$
where $P_{av}=\left(P_{0}+P_{1}\right)/2$ represents the average transmit power.

### 2.3. Atmospheric Turbulence Model

In order to model the intensity fluctuations caused by the atmospheric turbulence, the statistical Gamma-Gamma distribution is here assumed because of its mathematical tractability and accuracy to characterize a wide variety of scenarios ranging from weak to strong turbulence. Thus, and following [21], the probability density function (pdf) of $h_{a}$ is written as
(5)$$f_{h_{a}}\left(h_{a}\right)=\frac{2{\left(\alpha \beta \right)}^{(\alpha +\beta )/2}}{\Gamma (\alpha )\Gamma (\beta )}h_{a}^{(\alpha +\beta )/2-1}K_{\alpha -\beta }\left(2{\left(\alpha \beta h_{a}\right)}^{1/2}\right),$$
where $K_{p}\left(x\right)$ is the modified Bessel function of the second kind, and Γ(x) is the Gamma function, with α representing the effective number of large-scale cells of the scattering process and with β denoting the effective number of small-scale cells. Namely, from [39], these latter parameters, α and β, can be obtained as
(6)$$\alpha ={\left[\exp\left(0.49\sigma_{R}^{2}{\left(1+1.11\sigma_{R}^{12/5}\right)}^{-7/6}\right)-1\right]}^{-1},$$
and
(7)$$\beta ={\left[\exp\left(0.51\sigma_{R}^{2}{\left(1+0.69\sigma_{R}^{12/5}\right)}^{-5/6}\right)-1\right]}^{-1},$$
respectively, where $\sigma_{R}^{2}$ is the Rytov variance which, for uplinks paths, is defined as [39]
(8)$$\sigma_{R}^{2}=2.25k^{7/6}{\left(Z-z_{0}\right)}^{5/6}\int_{z_{0}}^{Z}C_{n}^{2}\left(z\right){\left(\frac{z-z_{0}}{Z-z_{0}}\right)}^{5/6}dz,$$
with $C_{n}^{2}\left(z\right)$ being the index of refraction structure parameter at altitude *z*, whereas k=2π/λ is the optical wave number and *Z* and $z_{0}$ are the UAV and transmitter heights, respectively.

### 2.4. Atmospheric Attenuation

For optical waves, the effect of the atmospheric attenuation suffered by the light propagating through the atmosphere is mainly caused by either absorption as scattering by air molecules in addition to both absorption and scattering by solid or liquid particles suspended in the air which, as a last resort, indicates the effect of weather conditions on the transmitted laser beam. Absorption and scattering are often grouped together under the topic of extinction, defined as the reduction or attenuation in the amount of radiation passing through the atmosphere. Mathematically speaking, such attenuation is incorporated using the well-known Beer–Lambert law [42,43,44] given by:(9)$$h_{l}=\exp\left(-a(\lambda )L\right),$$
where a(λ) is the wavelength dependent attenuation coefficient (extinction coefficient) and $L=Z-z_{0}$ is the propagation path length from the transmitter. As commented, the coefficient *a* depends on weather conditions and can be obtained from the atmospheric visibility. Since the attenuation coefficient hardly changes over long periods of time, we have assumed $h_{l}$ as a deterministic coefficient in our analysis.

### 2.5. Geometric Spread and Pointing Error Model

In addition to attenuation and atmospheric turbulence, geometric beam spread and pointing accuracy also affect the performance of these systems. The geometric loss is caused by the divergence of the transmit beam when propagating through the atmosphere, as $\omega_{z}=\theta_{T}\cdot L$, with $\omega_{z}$ being the received beam waist, with $\theta_{T}$ denoting the transmitter divergence angle, whereas *L* is the propagation path length. Since the received beam width is usually wider than the lens aperture size, part of the transmitted power cannot be collected, leading to loss. Thus, this geometric loss depends mainly on the ratio between the received beam waist, $\omega_{z}$, and the receiver aperture radius, *r*. However, during the designing of a FSO link, it is possible to control the beamwidth produced at a certain distance, and therefore the ratio $\omega_{z}/r$, by adjusting properly the laser parameters.

On the other hand, imperfections in the pointing, acquisition, and tracking process between the ground stations and the UAVs can also cause loss. Note that both phenomena are interrelated and, thus, accurate modeling frameworks have been proposed to account for both effects. To this end, in this work, we consider the general model proposed in [44,45]. According to this model, a Gaussian beam profile is assumed with a beam waist, $\omega_{z}$, on the receiver plane and a circular aperture receiver of radius *r*; then, the attenuation due to the geometric spread with pointing error can be approximated as the Gaussian form
(10)$$h_{p}\approx A_{o}\exp\left(-\frac{2\rho^{2}}{\omega_{zeq}^{2}}\right),$$
where ρ, is the radial pointing error, $A_{o}$ is the fraction of collected power without pointing error, i.e., only due to geometric spread, and $\omega_{zeq}^{2}$ is the equivalent beam width. Here, $A_{o}$ and $\omega_{zeq}^{2}$ are given by $A_{o}=\mathrm{erf}{\left(\mu \right)}^{2}$ and $\omega_{zeq}^{2}=\omega_{z}^{2}\sqrt{\pi }\mathrm{erf}\left(\mu \right)/\left[2\mu \exp\left(-\mu^{2}\right)\right]$, respectively, being erf(·) the error function and $\mu =\sqrt{\pi }r/\left(\sqrt{2}\omega_{z}\right)$. Moreover, considering independent identical Gaussian distribution for the horizontal *x* and *y* displacement in the receiver plane, the radial error $\rho =\sqrt{x^{2}+y^{2}}$ is modeled as Rayleigh distribution with a jitter variance at the receiver $\sigma_{s}^{2}$. Under these assumptions, the channel attenuation, $h_{p}$, can be seen as a function of the radial displacement, ρ, which is a RV. Hence, the pdf of $h_{p}$ can be seen as a random variable transformation problem, which leads to the following expression [45]:(11)$$f_{h_{p}}\left(h_{p}\right)=\frac{\gamma^{2}}{A_{o}^{\gamma^{2}}}h_{p}^{\gamma^{2}-1},$$
where $\gamma =\omega_{zeq}/\left(2\sigma_{s}\right)$ denotes the ratio between the equivalent beam radius at the receiver and the pointing error displacement standard deviation.

### 2.6. Composite Channel Model

Once the three factors included in Equation (Equation 3) have been individually discussed, the statistical characterization of the composite channel $h=h_{a}h_{l}h_{p}$ can be achieved. Thus, from [45],
(12)$$f_{h}\left(h\right)=\frac{2\gamma^{2}{\left(\alpha \beta \right)}^{\frac{\alpha +\beta }{2}}}{{\left(A_{o}h_{l}\right)}^{\gamma^{2}}\Gamma \left(\alpha \right)\Gamma \left(\beta \right)}h^{\gamma^{2}-1}\int_{\frac{h}{A_{o}h_{l}}}^{\infty }h_{a}^{\frac{\alpha +\beta }{2}-1-\gamma^{2}}K_{\alpha -\beta }\left(2\sqrt{\alpha \beta h_{a}}\right)\;dh_{a}.$$
where α and β parameters include the information on the strength of the turbulence, with γ containing the severity of the pointing error, where $A_{o}$ denotes the geometric spread attenuation and with $h_{l}$ being the path loss.

## 3. System Performance

In this paper, two metrics are considered to evaluate the performance of FSO vertical links. First, the amount of harvested energy in the UAV is analyzed through the average EH parameter. Second, the received signal quality is studied in terms of the bit error rate (BER) parameter.

In particular, we must remark that the harvested energy analysis detailed below is valid for any modulation scheme; nevertheless, our BER analysis will be particularized for a variant of the OOK scheme previously optimized for energy harvesting under a fixed peak power constraint [31]. An OOK scheme was selected because of its simplicity and low power consumption, especially interesting due to limited on-board UAVs battery lifetime. In such a scheme, named OOK-EH, a power $P_{1}=P_{m}$ is emitted for the transmission of a logical ‘1’, where $P_{m}$ is the peak power of the laser; whilst a fraction of $P_{m}$ is carried for transmitting a logical ‘0’, i.e.,
(13)$$P_{0}=\zeta P_{m},$$
where 0≤ζ<1. In contrast to the classical OOK, where no light is emitted when a logical ‘0’ is transmitted, our proposed scheme always carries a power level that will depend on the ζ parameter. This added DC component increases the average optical power $P_{av}$ and can be used to improve energy collection by adjusting the ζ parameter appropriately. Furthermore, and in order to maximize the harvested energy, a power level $P_{m}$ is also emitted when the transmitter is idle. However, any increase in the $P_{0}$ level will reduce the dynamic range given by the peak average optical power ratio (PAOPR) and, consequently, its associated BER will be increased. The PAOPR is given by 2/(1+ζ). Consequently, there is an important trade-off between increasing harvested energy and degrading BER. For any particular set of channel conditions, this trade-off is achieved for an optimal ζ that maximizes the harvested energy while keeping the BER performance below a predetermined target BER.

In order to evaluate the performance of any vertical FSO link with our proposed OOK-EH technique under realistic channel conditions, analytical closed-form expressions for the average EH and BER are derived in this section, considering the aforementioned main phenomena degrading the quality of the received signal, i.e., noise, turbulence, path loss, geometric spread and pointing errors.

### 3.1. EH Performance

An EH module is added to the receiver to collect the energy from the received signal. This module extracts the DC component, $I_{\mathrm{DC}}$, of the electrical signal that is obtained by the PD, *i*. The DC current is then either stored or directly used to feed other modules such as the information detection module and the UAV’s motor, which translates into extending the UAV’s battery lifetime. Figure 2 illustrates a block diagram of the receiver where the EH and information detection modules are shown. As it can be observed, the EH module is formed by a Schottky diode and a low-pass filter (LPF), which are passive devices, and thus, its associated power consumption is limited and could be included in the conversion efficiency, ζ [46]. Furthermore, this approach does not require either any control logic or any additional module to obtain the DC current.

As described in [29,31,47], the harvested energy per second in the optical receiver is given by
(14)$$EH=fV_{t}I_{DC}\ln\left(1+\frac{I_{DC}}{I_{o}}\right),$$
where *f*, $V_{t}$ and $I_{o}$ stand for the photo-detector’s fill factor, thermal voltage, and dark saturation current, respectively. The DC component of the output current, $I_{\mathrm{DC}}$, is written as [47]:(15)$$I_{DC}=RhP_{av}.$$

Here, *R* denotes the PD responsivity in A/W, with *h* being the composite channel attenuation coefficient, whereas $P_{av}$ represents the average transmitted power.

As can be noticed, the harvested energy is random due to the influence of the channel attenuation coefficient *h*. Therefore, to obtain the collected energy over a long period of time, the average EH (AEH) is derived as:(16)$$\mathrm{AEH}=\int_{h>0}f_{h}\left(h\right)fV_{t}RhP_{av}\ln\left(1+\frac{RhP_{av}}{I_{o}}\right)dh,$$
where $f_{h}\left(h\right)$ is the pdf of the composite channel attenuation according to (Equation 12). Employing the expressions [48] (Eqs. (07.34.21.0011.01) and (07.34.21.0085.01)), the solution of this integral can be written in a closed-form as
(17)$$\mathrm{AEH}=\frac{\gamma^{2}fV_{t}RP_{av}A_{o}h_{l}}{\alpha \beta \Gamma \left(\alpha \right)\Gamma \left(\beta \right)}G_{5,3}^{1,5}\left(\frac{A_{o}h_{l}RP_{av}}{I_{o}\alpha \beta }|\begin{array}{c} 1,1,-\gamma^{2},-\alpha ,-\beta \\ 1,-1-\gamma^{2},0 \end{array}\right).$$

Moreover, when only the effect of turbulent fading is considered, and $f_{h}\left(h\right)$ is given by (Equation 5), the above equation reduces to
(18)$$\mathrm{AEH}=\frac{fV_{t}RP_{av}{\left(A_{o}h_{l}\right)}^{2}}{{\left(\alpha \beta \right)}^{2}\Gamma \left(\alpha \right)\Gamma \left(\beta \right)}G_{5,3}^{1,5}\left(\frac{A_{o}h_{l}RP_{av}}{I_{o}\alpha \beta }|\begin{array}{c} 1,1,-1,-1-\alpha ,-1-\beta \\ 1,-2,0 \end{array}\right).$$
where the term $P_{av}$, which is present in Equations (Equation 16)–(Equation 18), is expressed as $P_{av}=$$P_{m}\left(1+\zeta \right)/2$ in case of OOK-EH. From Equations (Equation 17) and (Equation 18), the amount of harvested energy is not depending on the modulation scheme itself but on the average transmitted power. Of course, the particular channel conditions and the type of PD and rest of modules shown in Figure 2 will influence on the amount of energy any UAV can extract.

### 3.2. EH Optimization

In view of the expressions for the average harvested energy derived above, it is straightforward to observe that they depend on: first, the channel and link conditions (i.e., attenuation and turbulence, fundamentally); second, on the transmitter and receiver systems with a given pointing misalignment loss, geometric spread, and maximum peak power, $P_{m}$; and third, on the parameter ζ. In this section, we consider all these parameters as uncontrollable variables that depend on the channel and the physical devices, except for the ζ variable, that can be adjusted to maximize the harvested energy.

Therefore, we notice that if ζ increases, then the average harvested energy also increases; however, this fact adversely affects on the BER performance of the system. In this respect, there exists an interesting trade-off between EH and reliability. We consider in this work that the communication system includes a simple forward error correction (FEC) decoder, which defines a maximum pre-FEC BER, normally referred to as ${\mathrm{BER}}_{\mathrm{target}}$. Such a target is the maximum BER of the coded bits before the FEC decoder to guarantee an error-free transmission with high probability.

We assume two widely adopted target BER values of ${\mathrm{BER}}_{\mathrm{target}}=$$5\times 10^{-5}$ and ${\mathrm{BER}}_{\mathrm{target}}=$$10^{-8}$ related to different FEC options as defined in [49]. Interestingly, the latter target BER is related to a BASE-R code, which is also know as Fire Code FEC (FC-FEC) as is defined in (clause 74 of [49]). This latter code requires a very small computational complexity for its decoding.

Thus, we pose the optimization problem for the design of the proposed FSO system with EH as follows
(19)$$\begin{array}{cc} \; & \arg\min_{\zeta \in [0,1)}\mathrm{AEH}\\ \; & s.t.\;\mathrm{BER}<{\mathrm{BER}}_{\mathrm{target}} \end{array}$$

The above problem is solved numerically in Section 4 to determine the optimal ζ value and assess the maximum AEH that can be obtained with the proposed framework under realistic and diverse channel conditions.

### 3.3. BER Performance

Now, we derive closed-form expressions of the BER for the OOK scheme used for SLIPT. Hence, the received BER can be written as
(20)$$\mathrm{BER}=p_{0}P_{e}\left(1|0\right)+p_{1}P_{e}\left(0|1\right),$$
where $P_{e}\left(1|0\right)$ and $P_{e}\left(0|1\right)$ represent the probability of false alarm and the probability of missed detection, respectively. In our analysis, we assume equally-likely symbols, $p_{0}=p_{1}=0.5$ in many different realistic scenarios.

First, for an ideal scenario without neither turbulence nor pointing error (in (Equation 3), *h* is reduced to $h=A_{o}h_{l}$), the terms $P_{e}\left(1|0\right)$ and $P_{e}\left(0|1\right)$ can be expressed as [50]: (21)$$\begin{array}{c} P_{e}\left(1|0\right)=\frac{1}{2}\mathrm{erfc}\left(\frac{i_{t}-i_{s_{0}}}{\sqrt{2\sigma_{n}^{2}}}\right)=\frac{1}{2}\mathrm{erfc}\left(\frac{i_{t}-i_{s_{1}}\zeta }{\sqrt{2\sigma_{n}^{2}}}\right) \end{array}$$(22)$$\begin{array}{c} P_{e}\left(0|1\right)=\frac{1}{2}\mathrm{erfc}\left(\frac{i_{s_{1}}-i_{t}}{\sqrt{2\sigma_{n}^{2}}}\right); \end{array}$$
with $i_{s0}$ and $i_{s1}$ denoting the received photocurrent corresponding to the transmission of a logical ‘0’ and a logical ‘1’, respectively; where $i_{t}$ is the optimal detection threshold and $\sigma_{n}^{2}$ is the noise variance given in Equation (Equation 2). On another note, and from (Equation 13), the photocurrents $i_{s0}$ and $i_{s1}$ can be expressed as $i_{s0}=Rh\zeta P_{m}$ and $i_{s1}=RhP_{m}$, respectively. In addition, $i_{t}$ can be written as a function of ζ as
(23)$$i_{t}=\frac{i_{s_{1}}+i_{s_{0}}}{2}=\frac{i_{s_{1}}\left(1+\zeta \right)}{2}.$$

Recall that, for this ideal AWGN channel, *h* has a deterministic behavior since neither turbulence-induced nor misalignment fadings were considered yet. Accordingly, its associated BER is obtained by substituting (Equation 21) and (Equation 22) into (Equation 20), and it is given by
(24)$$\mathrm{BER}=\mathrm{erfc}\left(\frac{RhP_{m}\left(\zeta -1\right)/2}{\sqrt{2\sigma_{n}^{2}}}\right).$$

Second, we consider a more realistic scenario where atmospheric turbulence is now considered, although misalignment fading is still not included Therefore, the channel attenuation coefficient, *h*, becomes a RV since the turbulence-induced scintillation now incorporated into *h* follows the Gamma-Gamma pdf shown in (Equation 5). Thus (Equation 24) must be averaged with the corresponding pdf describing the behavior of *h*, and [48] (Eqs. (07.34.21.0013.01) and (07.34.21.0085.01)) are, again, used to solve the resulting integral:


(25)
$$\mathrm{BER}=\frac{2^{\left(\alpha +\beta \right)}}{8\pi \sqrt{\pi }\Gamma \left(\alpha \right)\Gamma \left(\beta \right)}G_{5,2}^{2,4}\left(\frac{8{\left(RA_{o}h_{l}P_{m}\left(\zeta -1\right)/2\right)}^{2}}{{\left(\sigma_{n}\alpha \beta \right)}^{2}}|\begin{array}{c} \frac{1-\alpha }{2},\frac{2-\alpha }{2},\frac{1-\beta }{2},\frac{2-\beta }{2},1\\ 0,\frac{1}{2} \end{array}\right)$$


Finally, when pointing errors are also taken into account, (Equation 21) and (Equation 22) must be now averaged with respect to (Equation 12). Hence,
(26)$$\begin{array}{c} \mathrm{BER}=\frac{2^{\left(\alpha +\beta \right)}\gamma^{2}}{16\pi \sqrt{\pi }\Gamma \left(\alpha \right)\Gamma \left(\beta \right)}\\ G_{7,4}^{2,6}\left(\frac{8{\left(RA_{o}h_{l}P_{m}\left(\zeta -1\right)/2\right)}^{2}}{{\left(\sigma_{n}\alpha \beta \right)}^{2}}|\begin{array}{c} \frac{1-\gamma^{2}}{2},\frac{2-\gamma^{2}}{2},\frac{1-\alpha }{2},\frac{2-\alpha }{2},\frac{1-\beta }{2},\frac{2-\beta }{2},1\\ 0,\frac{1}{2},\frac{-\gamma^{2}}{2},\frac{1-\gamma^{2}}{2} \end{array}\right), \end{array}$$
after having used [48] (Eqs. (07.34.21.0013.01) and (07.34.21.0085.01)). As a previous step, either the complementary error function, (erfc(·)), and the Bessel’s K function were expressed in terms of Meijer’s G functions from [48] (Eqs. (03.04.26.0006.01) y (06.27.26.0006.01)).

As can be seen from these latter equations, the exact expression for the error probability is given in terms of Meijer’s G-function, which may be difficult to facilitate further analytical studies. Hence, it would be possible to obtain simpler expressions after some mathematical approximations following the approximation given in [51] involving an upper bound and a lower bound for the Gaussian Q-function that represents the behavior in terms of error probability of an ideal Gaussian channel, as shown in Equation (Equation 22), and based on series expansion. Additionally, and for a future work, we have planned to use a more generic model for the turbulence, the Málaga model [52] and its formulation from Generalized-K functions [53]. Thus, as commented in [54], its pdf can be approximated by a Gauss–Laguerre quadrature. Since the Gamma-Gamma model employed in our paper is a particular case of the Málaga model [52], then this Gauss–Laguerre quadrature can also be applied. Thus the upper bound given in [54] and [51] (Equation (Equation 19)) can be employed.

## 4. Results and Discussions

In this section we analyze the impact of FSO link parameters on system performance in terms of energy harvesting and quality using the AEH and BER expressions derived in the previous section. This analysis allows to identify which are the key design parameters and the trade-offs to be considered in order to properly choose such parameters.

For our numerical analysis, we have assumed the values given in Table 1. In particular, we consider a vertical FSO link consisting of a ground-based optical transmitter located at a height $z_{0}$ and a receiver mounted on a UAV hovering on the ground at a height *Z*. Therefore, the separation between both is $L=Z-z_{0}$. The transmitter transmits a light beam with a divergence angle $\theta_{T}$= 1 mrad [6] and a peak power $P_{m}$ operating at a wavelength of 1550 nm. The receiver, as in [44], is composed of an optical circular converging lens whose effective light collection area is characterized by an aperture radius, *r*, of 10 cm with a responsivity R=0.5 A/W. Some commercial implementations following these features can be found at [55,56] Therefore, for L=300 m [57], the ratio of the beam waist to the aperture radius of the receiver is $(\omega_{z}/r)=3$ (remember that a source with a 1 mrad divergence angle was considered) while, for L=1000 m, $(\omega_{z}/r)=10$. To cover a large area on the ground, a field of view FOV $=45^{\circ }$ is assumed. Due to this large FOV, shot noise caused by ambient light is the dominant source of noise in the receiver. A spectral radiance $S_{n}=1$ mW/(cm${}^{2}$ nm srad), with an optical bandwidth $B_{o}=10$ nm and a noise bandwidth $B_{n}=1$ GHz are assumed to obtain $\sigma_{n}^{2}$ from (Equation 2).

Regarding the turbulence, we calculate the magnitudes of α and β, with the expressions (Equation 6) and (Equation 7), considering the values of $C_{n}^{2}\left(z\right)$ provided by the Hufnagel–Valley model [39] for different link heights, assuming $C_{n}^{2}\left(z_{0}\right)=1.7\times 10^{-13}$ m${}^{-2/3}$. Thus, α and β vary between 25 and 30 for heights between 300 m and 1000 m, leading to $\sigma_{R}^{2}<<1$, which corresponds to a very weak turbulence condition. In addition, we consider a jitter of $\sigma_{s}=10$ cm [58] to model the pointing error.

Finally, we use the attenuation coefficients corresponding to very clear air, clear air, and haze shown in Table 1.

Figure 3 shows the typical attenuation of vertical optical links (Figure 3a) together with the potential energies harvested (Figure 3b) for different separation distances between any UAV and the ground BS, ranging between 200 m and 1000 m.

Channel loss depicted in Figure 3a is calculated by $-10\log(\bar{h})$, with $\bar{h}=\left(h_{l}A_{o}\gamma^{2}\right)/\left(1+\gamma^{2}\right)$ representing the average attenuation coefficient calculated from Equation (Equation 12). It is straightforward to check how the coverage provided by any deployed UAV enhances with increasing its altitude but, as shown in Figure 3a, at the cost of a significant increase in power losses at the receiver. Specifically, that increase is mainly caused by the beam broadening induced by the transmitter divergence. Namely, for L=300 m (i.e., $(\omega_{z}/r)=$3, as commented above), a power loss of 8.5 dB is reached for $\sigma_{s}$ = 0.1 m. That value grows to 18 dB for L=1000 m as $(\omega_{z}/r)=$10. In both cases, a transmission divergence of 1 mrad and a 10 cm aperture radius were considered.

In addition, Figure 3b shows that increasing the channel loss dramatically reduces the harvested energy. Thus, for a single transmitter with a $P_{av}=500$ mW, the AEH drops from 18 mJ/s for the best case at 200 m; to less than 2 mJ/s at 1000 m. Note that the AEH in the figure has been calculated using Equation (Equation 17). Finally, Figure 3 shows how atmospheric losses are not relevant for the amount of energy the UAV can harvest due to the short link lengths considered in the analysis. Furthermore, they can be neglected without loss of accuracy when the UAV is affected by UAV’s jitter caused by its motor vibration.

Moreover, both figures also show the impact on the link performance of the pointing deviation caused by the jitter inherent to the UAV that acts as receiving side. For example, it can be observed that an increase in the UAV’s jitter (higher $\sigma_{s}$) is more detrimental when flying at lower altitudes than at higher altitudes. The reason for this is that the $\omega_{z}/r$ ratio decreases for lower altitudes due to the beam divergence angle and, thus, the random variations inherent to the UAV location cause a more serious channel loss than in higher altitudes, even causing that the communication link may even be interrupted if the UAV moves out of the transmitted beam footprint. Consequently, energy harvesting is less affected by jitter as the UAV is operating at a higher altitude and, accordingly, the $\omega_{z}/r$ ratio increases. Figure 4 summarizes this discussion.

Following the analysis of the channel behavior from Figure 3, now, Figure 5 shows the BER performance of the OOK and OOK-EH schemes under different channel impairments as a function of the received SNR. To illustrate the behavior of the OOK-EH scheme, a ζ=0.8 has been assumed. In addition, two ratios $\omega_{z}/r$ have been taken as representative values: $(\omega_{z}/r)=$ 3 and $(\omega_{z}/r)=$10, corresponding to UAV heights of 300 m and 1000 m, assuming a transmission divergence $\theta_{T}=$ 1 mrad. BER curves plotted in black represent the case of $(\omega_{z}/r)=$ 3; whilst the curves plotted in red depict the performance for $(\omega_{z}/r)=$ 10. In addition, solid lines represent the BER of the ideal channel with $h=A_{o}$, i.e., assuming only the geometric loss; whereas the dashed lines include the adverse effect of the random medium, i.e., either solely turbulence or both turbulence and pointing error.

These results, which are calculated from the expression (Equation 25) for the case of only considering turbulence-induced fading; and from (Equation 26) for the combined scenario with turbulence and pointing error, have been validated using Monte Carlo simulations. Note that BER curves for the ideal channel and the one associated to the turbulent channel with no pointing errors are identical for both $\left(\omega_{z}/r\right)$ ratios. In this figure, simulation (Monte Carlo) results are drawn with markers whereas theoretical results are drawn with either solid or dashed lines. In all cases, a perfect match is shown between the simulated results and those obtained from the derived expressions.

From the aforementioned Figure 5, it can be seen that turbulence and pointing error affect the BER with different severity depending on the $\omega_{z}/r$ ratio. As far as turbulence is concerned, it affects BER equally regardless of the ratio $\omega_{z}/r$. However, as expected, pointing error causes a very severe degradation for lower $\omega_{z}/r$ ratios. The figure shows that, for $(\omega_{z}/r)=$ 3, the SNR degradation caused by pointing errors is huge, on the order of 15 dB for a target BER of $5\times 10^{-5}$, while for $(\omega_{z}/r)=$ 10, the degradation is nearly negligible. In fact, the BER curves considering turbulence and turbulence with pointing error almost overlap.

### Optimization for EH

As described in the previous section, the process of optimizing the modified OOK scheme for EH consists of choosing the optimal ζ value that maximizes the average EH while keeping the BER below the target value for each channel condition. Note that, since the increase in EH is achieved at the cost of degrading the link quality, the maximum ζ value will depend on the considered channel impairments, i.e., turbulent fading, pointing error, and atmospheric attenuation. Consequently, a lower channel degradation will lead to higher ζ values and, thus, to higher harvested energies. The value of the optimal ζ has been obtained numerically from expression (Equation 26) by setting the link parameters and the target BER.

To form a complete picture of the dependence of optimum ζ on the FSO link parameters, Figure 6 shows the optimal value of ζ as a function of transmitter peak power considering different $\omega_{z}/r$ ratios, jitters and target BERs. In all cases, the turbulence fading and the path loss corresponding to “very clear air” are included. As can be seen in the figure, as the peak power of the transmitter increases, the optimum ζ value also increases. In our analysis, realistic power values up to 35 dBm have been considered. Note that, since the UAV is intended to collect as much energy as possible, the power value chosen in the link design will be high. Therefore, the optimal ζ values will also be high. For the highest of the powers considered ($P_{m}=$35 dBm), the value of ζ is always higher than 0.8. In addition, as expected, the optimal ζ values obtained for the $10^{-8}$ target rate (red lines) are lower than for $5\times 10^{-5}$ (blue lines) and, similarly, higher jitter values lead to lower ζ values. From that Figure 6, it can be concluded that values of ζ higher than 0.8 one can be selected for realistic propagation scenarios.

On a different matter, Figure 7 depicts the average harvested energy for the OOK and OOK-EH schemes as a function of the peak transmitted power. For the OOK-EH scheme, the optimal ζ values obtained in Figure 6 have been here employed.

Hence, from the AEH results depicted in Figure 7, the following comments can be drawn. First, it is clearly observed that the energy collected with the OOK-EH scheme (black lines) is higher than that of the OOK scheme (red lines). In fact, the energy harvested by the OOK-EH scheme tends to twice that in OOK scheme as the transmitted power increases. Figure 7 also shows that a minimum $P_{m}$ is required to achieve the previously selected BER target (a power below $P_{m}$ cannot satisfy the performance required by the BER target and, accordingly, no energy would be harvested). This $P_{m}$ is higher for the ratio $\omega_{z}/r=$3 than for $\omega_{z}/r=$10 due to what was explained when introducing Figure 3, Figure 4, Figure 5 and Figure 6. Therefore, the choice of the ratio $\omega_{z}/r$ has a huge impact on the value of the collected energy. The figure shows that the energy obtained with $\omega_{z}/r=3$ is much higher than that obtained with $\omega_{z}/r=$10. In particular, for the highest power considered (Pm=35 dB), the AEH is about 70 mJ/s for $\omega_{z}/r=$3, while it hardly reaches 10 mJ/s for $\omega_{z}/r=10$. Note that the AEH values shown in that figure are consistent with those ones published by other authors in [31] and [47] for peak transmitted powers of 100 and 200 mW, respectively.

As explained above, one of the main features affecting energy harvesting is the $\omega_{z}/r$ ratio. In this respect, it is possible to design transmitters for EH with an accurate control of their beam width since significant gains can be achieved with the appropriate selection of $\omega_{z}/r$. Thus, when increasing this ratio, the pointing problem is relaxed and, as well, the negative effect inherent to the jitter is reduced, i.e., when increasing $\omega_{z}/r$ then fading induced by jitter suffered by the receiver is less intense since the received beam waist becomes (much) bigger than the receiver aperture radius. In this respect, although the received spot can wander due to the jitter effect, however, the total amount of caught power in the receiver side is maintained without significant variations since the size of the received beam waist is large compared to the receiver aperture radius. Consequently, the amount of power that can be captured from the section of the beam illuminating the receiver is, more or less, of the same magnitude. On the contrary, for smaller values of $\omega_{z}/r$, the sway in the received beam footprint caused by misalignment is more critical in the sense that such a sway may make all the received beam spot drop out of the receiver photosensitive area. For that critical situation, the amount of captured power in the receiver would drop to zero. Of course, large $\omega_{z}/r$ values lead to both severe geometrical losses (since most of the received footprint area is spread out of the physical photosensitive area implemented in the receiver side) and, consequently, low values of harvested energy. Therefore, the design of any FSO link should try to minimize this ratio by using adaptive pointing tracking systems [18].

It is worth noting that all the the results of average EH shown so far are for a single vertical FSO link between a given BS and UAV. Nevertheless, each UAV could receive simultaneous transmissions from a number of BSs as long as their locations fall within the region, $R\subset R^{2}$, covered by the FOV of the UAV’s receiver. The area of such a region, |R|, can be expressed in terms of the FOV angle and the height of the UAV, *Z*, as follows: $\left|R\right|=\pi {\left(Z\tan(\mathrm{FOV}/2)\right)}^{2}$. Therefore, the average number of BSs, *N*, that are covered by the FOV of the receiver is written as $N=\left|R\right|\lambda_{\mathrm{BS}}$, where $\lambda_{\mathrm{BS}}$ is the BS density expressed in number of nodes per $m^{2}$. Assuming that the macro BSs are spatially distributed according to a hexagonal grid, with a number of RRHs, $n_{\mathrm{RRHs}}$, randomly distributed within each macro cell as described in [59], that BS density leads to $\lambda_{\mathrm{BS}}=\frac{\left(1+n_{\mathrm{RRHs}}\right)}{2\sqrt{3}{\left(\frac{\mathrm{ISD}}{2}\right)}^{2}}$, where the term ISD stands for the inter-site distance (ISD), which is the distance between two neighbouring macro BSs. Finally, according to sections 6.1.2 and 6.1.4 of [59], Thus, for a given FOV, the value of *N* can be obtained as a function of the UAV height. In the case of dense urban microcellular deployment scenarios, the value of *N* increases from 6 to 62 when the height of the UAV increases from 300 m to 1000 m. However, despite the significant increase in the number of BS, the total collected energy hardly changes due to the increase of optical link loss with UAV height.

As a conclusion, considering the AEH results shown in Figure 7 and the number of BSs covered by the UAV’s receiver described above, the total AEH that can be obtained for considered scenario in very clear air conditions with a target BER of $5\times 10^{-5}$ and assuming realistic transmitted powers between 30 and 35 dBm is in the range between 134 and 450 mJ/s for a UAV height of 300 m and between 164 and 558 mJ/s for a UAV height of 1000 m. This free energy collected from the information-carrying FSO signals complements the energy harvested from the terrestrial optical and RF wireless power transfer (WPT) charging stations and contributes to extend the battery life of UAVs.

## 5. Concluding Remarks

In this paper, we have investigated the application of FSO, UAVs, and EH as an adaptable and efficient solution to provide backhaul/fronthaul connectivity to 5G+ networks. There are many benefits supporting this approach. First, FSO technology provides high bandwidth for the 5G RAN. Second, FSO communication links do not interfere with the RF based 5G RAN. Third, the UAVs, which act as flying nodes of a backhaul/fronthaul network, can adapt to changes in weather and traffic conditions to provide reliable links. Nevertheless, the limited battery of the UAVs causes service interruptions when the UAVs need to recharge them. For this reason, we propose the use of EH to collect energy from the transmission of information signals to combine with the EH from the terrestrial optical and RF WPT charging stations. All these techniques are thought to enhance the on-board battery lifetime of UAVs. To assess the benefits of the proposed approach we have considered a realistic yet tractable channel model that includes the effect of turbulence fading, pointing errors and atmospheric attenuation. Using this model, analytical closed-form expressions of the average harvested energy and the bit error rate of an OOK scheme optimized for information transmission and power transfer are derived. The derived expressions allow to evaluate the performance of vertical FSO links between ground-based BSs and UAVs and to properly select the link parameter to optimize the harvested energy while guaranteeing a reliable connection.

Results show that there exists an interesting trade-off between reliability and harvested energy.

## Figures and Tables

**Figure 1 sensors-22-05684-f001:**
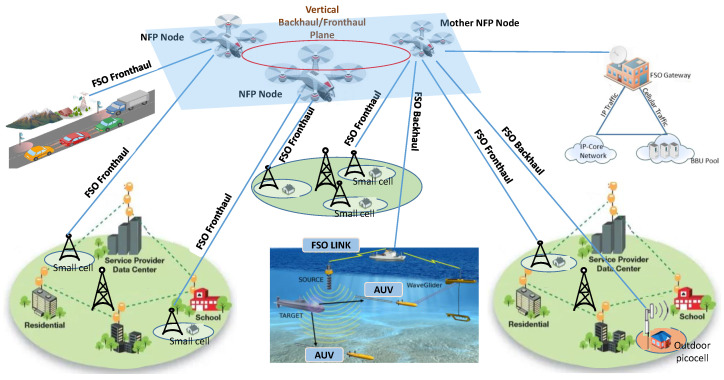
Possible 5G + network scenarios where FSO will extend or complement existing deployments.

**Figure 2 sensors-22-05684-f002:**
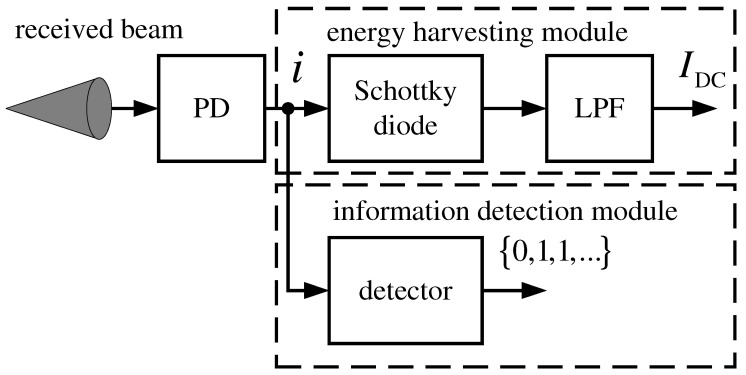
Block diagram of the proposed receiver structure with EH. An optical circular converging lens would be placed in front of the PD.

**Figure 3 sensors-22-05684-f003:**
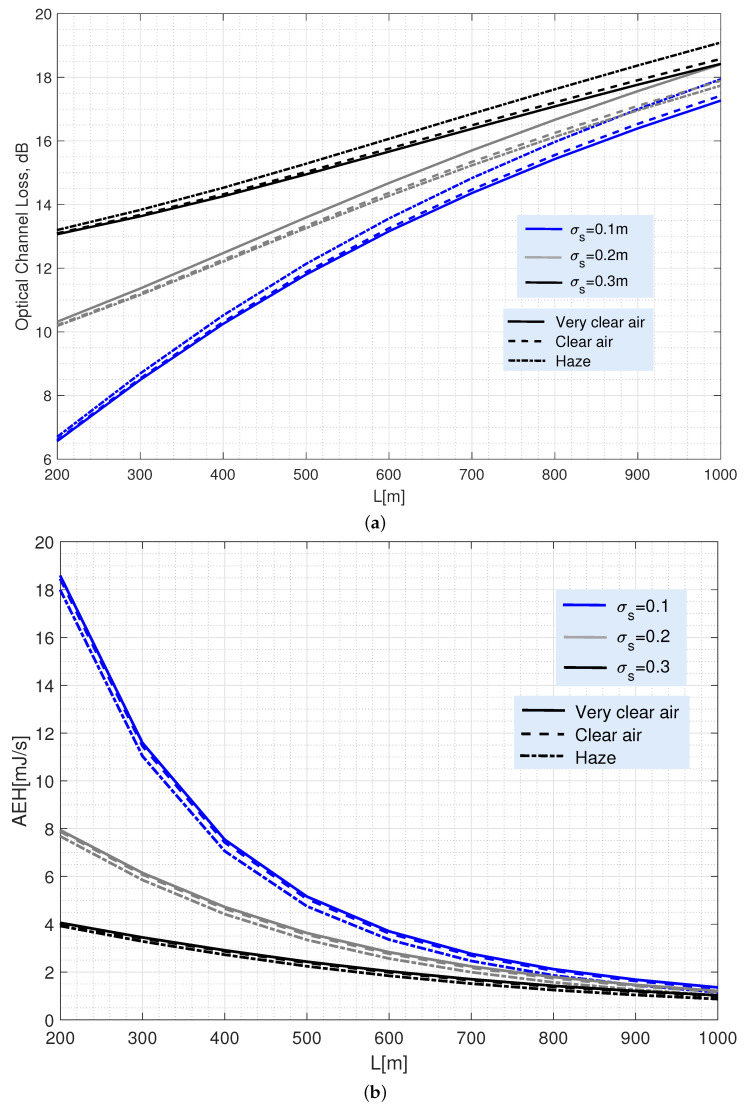
(**a**) Average optical channel loss as a function of link length. (**b**) Average harvested energy as a function of the link length. Both figures consider different UAV jitters and weather conditions and a beam divergence of 1 mrad. To obtain the AEH, a single transmitter with $P_{av}=$500 mW is assumed.

**Figure 4 sensors-22-05684-f004:**
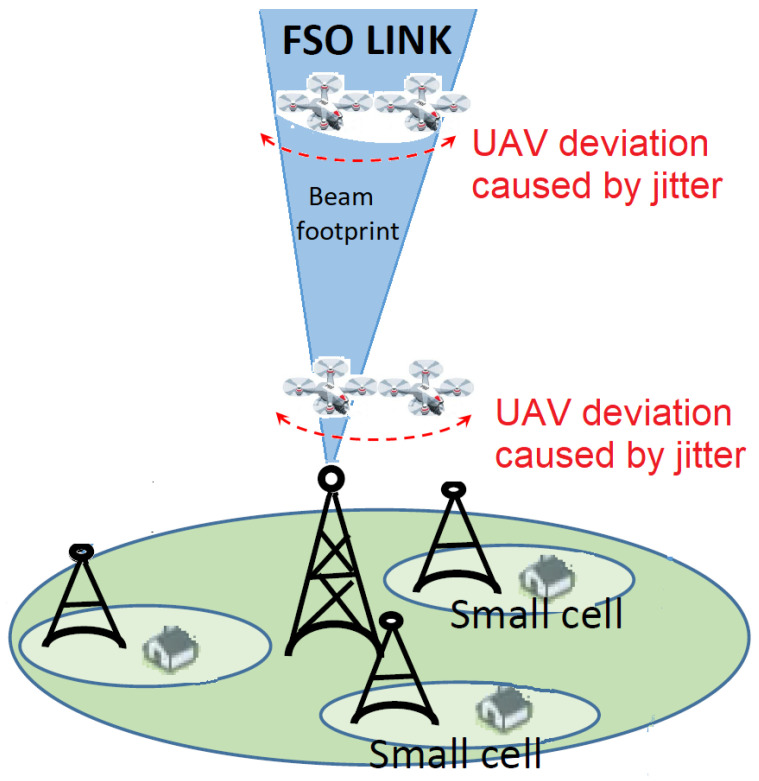
Adverse effect of UAV’s jitter caused by its motor vibration versus altitude in the flight state. It is supposed that the same UAV is flying at two different altitudes and affected by a same value of $\sigma_{s}$. For the case of the UAV operating at the lower altitude, the communication link may even be interrupted due to the presence of UAV’s jitter. Of course, for a higher altitude, the beam broadening induces more serious power losses at the receiver.

**Figure 5 sensors-22-05684-f005:**
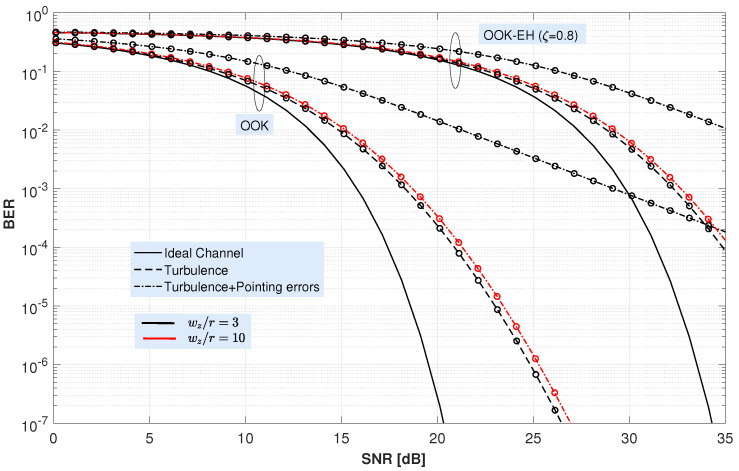
Average BER for the OOK and OOK-EH schemes versus signal-to-noise ratio for a vertical FSO link under different channel impairments and different values of the ratio $\left(w_{z}/r\right)$. A transmitter with a divergence $\theta_{T}=$1 mrad and a $P_{m}$ up to 20 dBm is assumed.

**Figure 6 sensors-22-05684-f006:**
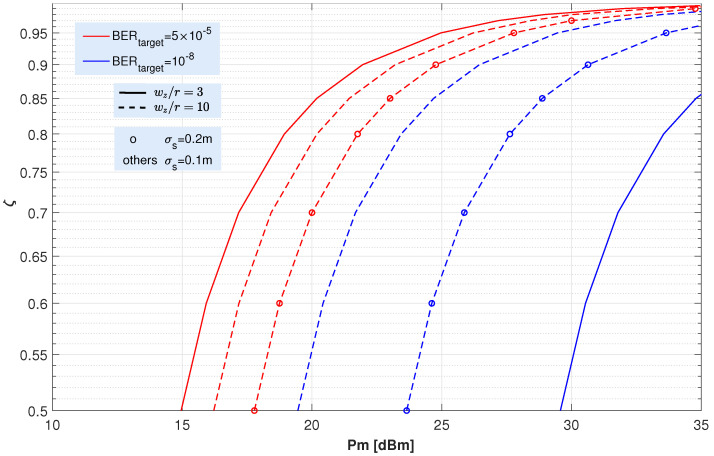
Optimal ζ values for the proposed OOK-EH scheme as a function of the transmitted peak power considering different $\left(\omega_{z}/r\right)$ ratios, jitters and target BER. These ζ values maximize the average harvested energy while maintaining the target BER. Turbulence fading (α=30, β=30) and “very clear air” conditions are assumed.

**Figure 7 sensors-22-05684-f007:**
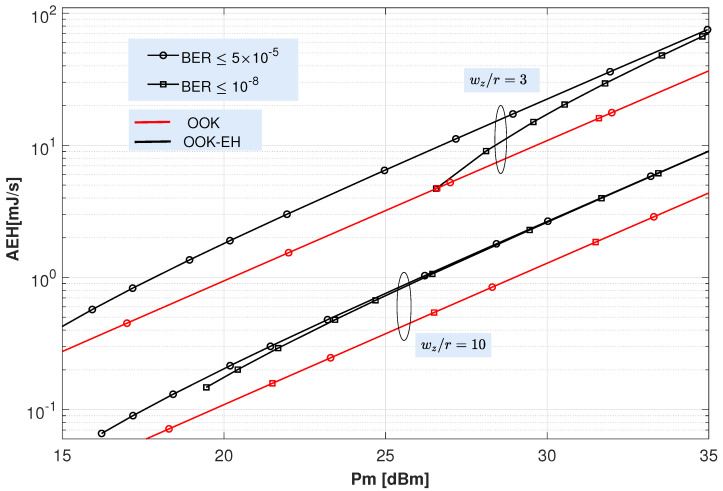
Average harvested energy for the OOK and OOK-EH schemes versus peak transmitted power considering different $\left(\omega_{z}/r\right)$ ratios and target BER. Turbulence fading (α=30, β=30), a jitter ($\sigma_{s}=0.1$ m) and “very clear air” conditions are assumed. A single transmitter has been considered.

**Table 1 sensors-22-05684-t001:** System parameters.

FSO Parameters		
**Parameter**	**Symbol**	**Value**
Operating Wavelength	λ	1550 nm
Transmitter divergence	$\theta_{T}$	1 mrad
Responsivity	*R*	0.5 A/W
Receiver aperture radius	*r*	10 cm
Field of view	FOV	$45^{\circ }$
Optical Bandwidth	$B_{o}$	10 nm
Noise Bandwidth	$B_{n}$	1 GHz
Spectral radiance	$S_{n}$	1 mW/cm${}^{2}$ nm srad
**Turbulence, pointing error and climatic parameters**		
**Parameter**	**Symbol**	**Value**
Structure parameter	$C_{n}^{2}\left(z_{0}\right)$	$1.7\times 10^{-13}$ m${}^{-2/3}$
Number of large-scale cells	α	30
Number of small-scale cells	β	30
Jitter variance	$\sigma_{s}$	10 cm
Attenuation coefficient (very clear air)	*a*	0.0647 dB/km
Attenuation coefficient (clear air)	*a*	0.2208 dB/km
Attenuation coefficient (haze)	*a*	0.7360 dB/km
**EH parameters**		
**Parameter**	**Symbol**	**Value**
Fill factor	*f*	0.75
Dark saturation current	$I_{o}$	$10^{-9}$ A
Thermal voltage	$V_{t}$	25 mV

## Data Availability

Not applicable.

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
