# Peer review of "Performance Analysis of a Vertical FSO Link with Energy Harvesting Strategy"

_sensors, 2022, doi:10.3390/s22155684_

Round 1

Reviewer 1 Report

This paper investigates the application of FSO, UAVs, and EH as an adaptable and efficient solution to provide backhaul/fronthaul connectivity to 5G+ networks. It is interesting that they considered a variant of the practical OOK modulation for information and power transfer. The mathematical derivations and numerical analysis are solid and complete. The paper is very well written. Only some minor points are addressed as follows:

1.     It is good to see that the results are derived in closed form and in terms of Meijer-G functions. But I wonder whether simpler expressions to facilitate further analysis can be obtained using some mathematical approximations. Please refer to the references below and discuss the differences.

Ref

[1] “Impacts of Pointing Errors on Error Performance of Inter-Satellite Laser Communications” in 2017 J. Lightw. Technol.

[2] “Tight Bounds and Invertible Average Error Probability Expressions over Composite Fading Channels” in 2016 J. Commun. And Networks.

2. The authors say that “Thus, when increasing this ratio, the pointing problem is relaxed and, as well, the negative effect inherent to the jitter is reduced (Why? I think the requirement on pointing system is higher for higher ratio). However, large w_z/r values lead to both severe geometrical losses and low values of harvested energy.” I think there is ambiguity between these two sentences. Please clarify and discuss more.

3. Some typos, e.g., “FNP”, “as as ”, and some wrong grammars, e.g., “Among them, the expressions for the BER of the system, the amount of harvested energy and the UAV power consumption model.” need to be revised.

Author Response

Attached (PDF file) you can find my point-by-point response to all your comments and suggestions. 
Thank you very much for your valuable comments.

Reviewer 2 Report

Free-space optical communication links (FSO) between the ground stations, UAVs and end-users are becoming a promising technology under the 5G+ strategy. It has also become a hot research topic to study the energy harvesting (EH) capability under the FSO technology, because basically, the DC part of the NRZ OOK signals can indeed be utilized to charge the UAV batteries. This work contributes to this field by investigating the BER characteristics of the FSO + EH system. The authors have come up with solid theoretical models, run sufficient simulations and verified their models using general Mente Carlo simulations. I believe this work is sound in theory and simulations.

However, my main problem is the feasibility of the proposed system: FSO + EH, as so far most of the works, including this one, are purely designs + simulations. One has yet to see some experiments to understand the true difficulty in the technology, especially in determining the parameters used in the design.

Specifically, I cannot recommend rapid publication at this stage for the following reasons.

1) The authors assume the receiver has an optical aperture of radius r = 10 cm. It must be pointed out that the typical receiver for 1550 nm is based on either InP or Si-Ge detector technology. A detector (array) of 20-cm in diameter, with a responsivity of 0.6 A/W at high data rates are extremely difficult to fabrication and assemble. The cost of the detector alone would render the whole UAV communication network impractical in real applications. The author should revise this parameter considering the (commercially) available detector choices, the cost and technical challenges.

2) Another far too optimistic parameter is the receiver beam waist (30 cm). Considering the laser source at 1550 nm can only be realized by the InP technology in the industry, the laser diode mostly have a beam waist of only a few micrometers. Indeed, an optical system can be built to the expand the beam waist / reduce the far-field radiation angle, but it has also limitation. Assuming a near-field beam waist of 1 mm and a propagation distance of 1000 m in air, the beam waist expands to the meter scale. The small collecting efficiency / huge loss would require a high power supply to the ground / base station. The authors should make a sound judgement whether such power consumption is practical for the transmitter station under their design scenarios and whether the proposed BER/RH compromise is still valid.

Author Response

(The authors gave the same response as above.)

Reviewer 3 Report

The article presents a theoretical study of a free-space optical link that is added by an energy-harvesting method. The contribution provides a couple of different aspects, which are all more or less known. The added value is meant to combine these aspects. 

Unfortunately, the article gives the impression of a list of things that do not much relate to each other. The actual point of the paper is hard to get, as there are many theoretical derivations on the optical channel and the linked bit-error probability. Most of the work is more or less known, but is extended by an energy-harvesting approach, by which the average transmitted power is intendedly increased by rising the zero-level of the transmission.

Very little explanations are given on the actual energy harvesting, e.g. on the aspect how much energy must be harvested. This would be interesting regarding the fact that target bit-error probabilities are chosen that require energy-greedy forward-error correction. Here, a less energy consuming transmission scheme could be more fortunate.

The final discussion of the results looks rather like an exemplary demonstration of the results that a systematic intend to find an optimum transmission scheme or prove that the given concept is actually feasible. Especially the lack of systematic derivation of system parameter, the involved compromises and the key performance parameters bring me to the conclusion that the paper could be of interest, but it not really focused and does not really provide new insights or novelty compared to the state of the art. 

The authors are encouraged to sharpen the actual goal of the paper, improve the methodology how to get there and to think about those explanations that might be skipped. I guess the main idea of the paper is a proof of concept that autonomous flying hops could be provided with energy and communication using this method. Then, the scenario should be better defined, the actual physical parameters (what are typical attenuations, how much outage probability can be allowed, what is the influence of the laser line width and the modulation format on things like irradiation variation) and methods (is on-off keying the best format for such a scheme? What kind of detector should be used? How much energy must be transmitted etc.) should be motivated.

The contribution in its current form is not really useful and provides mainly a collection of useful facts, but not really an answer to an interesting research question.

Author Response

(The authors gave the same response as above.)

Round 2

Reviewer 2 Report

The authors have addressed my questions well. I recommend it to be accepted.

Author Response

Thank you very much for your final recommendation.

Reviewer 3 Report

The authors have addressed all raised concerns and improved the article to an extent that it might be published. Thus, I looked more at the details and have some (hopefully) minor points that must be mended/commented on before publication:

1. It is still a bit confusing the the authors argue with theta as most important design parameter, but the closed-form expressions for harvested energy Eqs. (16-18) do not explicitly state the theta dependence. 

2. In 3.1 it is stated that the passive nature of the EH module would directly lead to negligible losses, which is surely not the case. For instance, an Ohmic resistor is purely passive, but dissipates all energy… Please rephrase this sentence. 

3. There is something strange about the Eq. (14). Eq. (14) described a power, not an energy as far as I understand the variables. This is also quite useful to do since this describes the harvested energy per time. But it should be named like a power or a harvested energy per second.

4. In Eqs. (16-18) the average harvested energy is calculated and later expressed in Joules. This must also be a power since it is just integrated over the losses and weighed by the probability density. Please adjust the naming and the units. 

5. The discussion is quite long and there is no obvious final result. This should be sharpened: What is the typical harvested energy that can be achieved? What are the conditions under which  this is achieved? Is this feasible with today’s technology and how does it compare to existing approaches? Please focus more on these aspects, which directly follow from the derivations within the article, and refrain more from potential other technologies that might be added to improve the energy harvesting. 

Author Response

Thank you for your comments. Attached you can find our detailed replay to your review report.
